# Experimental cheat-sensitive quantum weak coin flipping

Simon Neves [1] ✉, Verena Yacoub[1], Ulysse Chabaud [2,3], Mathieu Bozzio [4] ✉, Iordanis Kerenidis[5] & Eleni Diamanti [1]

As in modern communication networks, the security of quantum networks will rely on complex cryptographic tasks that are based on a handful of fundamental primitives. Weak coin flipping (WCF) is a significant such primitive which allows two mistrustful parties to agree on a random bit while they favor opposite outcomes. Remarkably, perfect information-theoretic security can be achieved in principle for quantum WCF. Here, we overcome conceptual and practical issues that have prevented the experimental demonstration of this primitive to date, and demonstrate how quantum resources can provide cheat sensitivity, whereby each party can detect a cheating opponent, and an honest party is never sanctioned. Such a property is not known to be classically achievable with information-theoretic security. Our experiment implements a refined, loss-tolerant version of a recently proposed theoretical protocol and exploits heralded single photons generated by spontaneous parametric down conversion, a carefully optimized linear optical interferometer including beam splitters with variable reflectivities and a fast optical switch for the verification step. High values of our protocol benchmarks are maintained for attenuation corresponding to several kilometers of telecom optical fiber.

Communication network users must operate or interact with potentially untrusted parties, servers, nodes and transmission channels in order to handle sensitive data, sign digitally, perform online banking, delegate computations, and electronically vote, among many other tasks. To guarantee the security of such network tasks against malicious entities, it is necessary to rely on a collection of building blocks, called cryptographic primitives, which can be combined with one another to guarantee overall security[1]. Coin flipping is a fundamental primitive that comes in two versions. In strong coin flipping (SCF), two parties remotely agree on a random bit such that none of the parties can bias the outcome with probability higher than $1/2 + \epsilon$, where $\epsilon$ is the protocol bias[2]. It is essential for multiparty computation[3], online gaming and more general randomized consensus protocols involving leader election[4]. In weak coin flipping (WCF), on the other hand, there is a winner and a loser, in the sense that both parties have a preferred, opposite outcome.

In classical communication networks, there exist no secure SCF and WCF protocols without computational assumptions or trusting a third party[2,5–7]. Although accepting a nonzero abort probability allows for information-theoretically secure classical schemes to exist[8], such schemes cannot detect malicious behaviours deviating from the original protocol. On the other hand, cheat-sensitive coin flipping becomes possible when using quantum properties. Quantum SCF protocols have in fact been shown to display a fundamental lower bound on their bias[9], but quantum WCF may achieve biases arbitrarily close to zero[10–12]. Interestingly, quantum WCF can also be used for the construction of optimal quantum SCF and quantum bit commitment schemes[13–15].

[1]Sorbonne Université, CNRS, LIP6, 4 Place Jussieu, Paris F-75005, France. [2]Institute for Quantum Information and Matter, California Institute of Technology, 1200 E California Blvd, Pasadena, CA 91125, USA. [3]DIENS, École Normale Supérieure, PSL University, CNRS, INRIA, 45 rue d'Ulm, Paris 75005, France. [4]University of Vienna, Faculty of Physics, Vienna Center for Quantum Science and Technology (VCQ), 1090 Vienna, Austria. [5]Université de Paris, CNRS, IRIF, 8 Place Aurélie Nemours, Paris 75013, France. ✉e-mail: simon.neves@laposte.net; mathieu.bozzio@univie.ac.at

While quantum SCF protocols have been experimentally demonstrated[16–18], along with other quantum two-party computations[19–23], the implementation of quantum WCF has remained elusive so far, because of the absence of protocols bringing together the use of practical states and measurements with tolerance to losses. Recently, a linear optical implementation, exploiting photon-number encoding, was proposed in[24], but the quantum advantage it can provide in terms of bias is very sensitive to losses: a dishonest party may always declare an abort when they are not satisfied with the outcome of the coin flip.

Here, we provide an experimental demonstration for quantum WCF. Our demonstration relies on the generation of heralded single photons by spontaneous parametric down conversion (SPDC), which are effectively entangled with the vacuum on a beam splitter of variable reflectivity. The outcome of the coin flip is then provided by the detection or absence of a photon. Our protocol is a refined version of the theoretical protocol from[24], which provides a new desirable property in the presence of losses that relates to cheat sensitivity rather than bias: by dropping the condition from[24] that both parties have equal probabilities of winning when cheating, our protocol allows them to detect whether their opponent is cheating during a verification step, and does not sanction an honest party, while retaining security in terms of bias. There are no known classical protocols that achieve such cheat sensitivity[25,26]. In order to emphasize the robustness of our protocol to losses, we show that it remains secure over an attenuation that corresponds to several kilometers of telecom optical fiber.

## Results
### Protocol
We first introduce our protocol for quantum weak coin flipping using a single photon, building on the protocol proposed in[24]. Our protocol accounts for potential losses and the detection of a cheating party (see Fig. 1 and Box 1). It ends with five mutually incompatible outcomes: Alice wins or is sanctioned, Bob wins or is sanctioned, or the protocol aborts. The protocol uses three beam splitters, whose reflectivities $x$, $y$, and $z$ are chosen in order to satisfy two conditions on these events. Firstly, the *fairness* condition, which states that Alice and Bob have equal winning probabilities when both of them are honest, i.e. $\mathbb{P}_h(\text{A. wins}) = \mathbb{P}_h(\text{B. wins})$, or

$$\mathbb{P}_h\big[(b, v_1, v_2) = (0, 1, 0)\big] = \mathbb{P}_h\big[(b, a) = (1, 0)\big]. \qquad (1)$$

Secondly, the *correctness* condition, which states that an honest party should never be sanctioned for cheating, i.e. $\mathbb{P}_h(\text{A. sanctioned}) = \mathbb{P}_h(\text{B. sanctioned}) = 0$, or

$$\mathbb{P}_h\big[(b, v_2) = (0, 1)\big] = \mathbb{P}_h\big[(b, a) = (1, 1)\big] = 0. \qquad (2)$$

Note that contrary to the previous protocol[24], we drop the *balancing* condition, which states that Alice and Bob should have equal probabilities of winning when using an optimal cheating strategy, as it cannot be satisfied together with the correctness condition in presence of experimental imperfections. Consequently, a practical balanced protocol would sanction an honest Alice for cheating, with non-zero probability. This impacts the cheat sensitivity, as one cannot trust the verification step if it sanctions honest parties (see Supplementary Note 1 for details on the protocol and the chosen conditions).

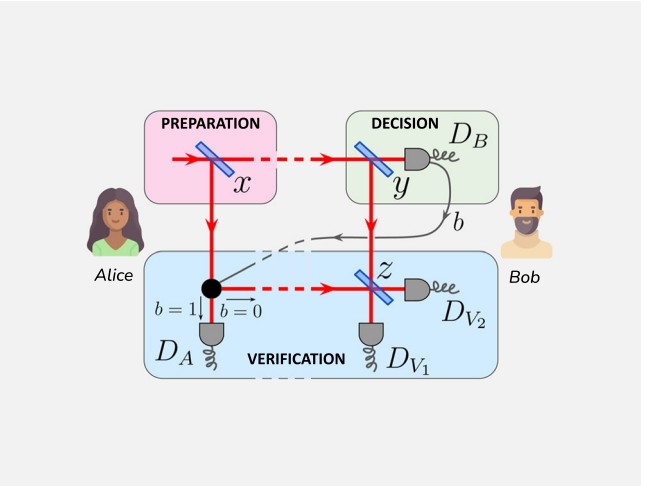

**Fig. 1 | Schematic description of cheat-sensitive quantum weak coin flipping.** The preparation, decision and verification steps, along with the role of measurement outcome $b$ and beam splitter reflectivities $x$, $y$, $z$, are detailed in Box 1. The sources for Alice and Bob representations can be found at Woman icons created by Freepik - Flaticon and Man icons created by Freepik - Flaticon, respectively.

---

## BOX 1

# Protocol for cheat-sensitive quantum weak coin flipping with a single photon

---

1. *Preparation*. Alice sends a single photon on a beam splitter of reflectivity $x$, keeps the reflected mode, and sends the other to Bob.

2. *Decision*. Bob sends the state he receives on a beam splitter of reflectivity $y$, measures the transmitted mode with a single-photon detector $D_B$, and broadcasts the outcome $b = 0$ (no photon detected) or $b = 1$ (a photon was detected).

3. *Verification*. If $b = 0$, Alice sends her reflected mode to Bob, who mixes it with his own reflected mode on a beam splitter of reflectivity $z$, and measures the two outputs with single-photon detectors $D_{V_1}$ and $D_{V_2}$. He distinguishes three cases depending on the corresponding outcomes $(v_1, v_2)$:

   - $v_2 = 1$: Alice is sanctioned for cheating,
   - $(v_1, v_2) = (1, 0)$: Alice wins,
   - $(v_1, v_2) = (0, 0)$: the protocol aborts.

   If $b = 1$, Bob discards his state. Alice measures her state with a single-photon detector $D_A$. She discerns two cases depending on the outcome $a$:
   - $a = 0$: Bob wins,
   - $a = 1$: Bob is sanctioned for cheating.

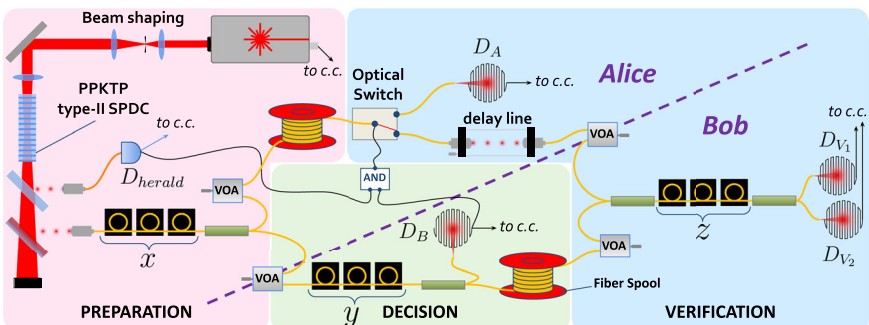

**Fig. 2 | Experimental setup for cheat-sensitive quantum weak coin flipping.** A ppKTP crystal (30 mm-long, 46.2 μm poling period) is pumped by a 770 nm pulsed laser (≈2 ps long pulses, 76 MHz rate). Twin photons at telecom wavelengths are generated via type-II SPDC, separated from the pump by a dichroic mirror (DM), and from each other by a PBS. The signal photon is used to perform the protocol as shown in the scheme in Box 1, which is the reference for defining the reflectivities $x$, $y$ and $z$. These are tuned with polarization controllers, placed before PBSs. At the end of each of the possible paths, the signal photon is detected with high-efficiency SNSPDs ($D_A$, $D_B$, $D_{V_1}$, $D_{V_2}$). A coincidence counter (c.c.) conditions each detection on the idler photon, detected with an InGaAs avalanche photodiode (APD) $D_{herald}$, and on the emission of a pump pulse, detected via an internal photodiode, measured in a 500 ps coincidence window. The signal from Bob's detector, conditioned on the heralding signal via a logic AND gate, triggers a fast optical switch on Alice's side. While these signals are being processed, the photon is delayed by 300 m-long optical fiber spools on each party's side. A delay line allows for fine-tuning of the wave-packets timing on the last PBS. Communication distance $L$ between Alice and Bob is simulated by VOAs of transmission $e^{-0.02L}$, which are shown on the dashed line marking visually the separation between the two parties. Two more VOAs are included in the setup to simulate losses due to photon storage corresponding to this distance.

## Experimental setup

The experimental setup used for the implementation of our protocol is shown in Fig. 2. Alice generates heralded single photons via type-II SPDC in a periodically-poled potassium titanyl phosphate (ppKTP) crystal. The protocol is implemented using fibered components at telecom wavelength. As polarization is a degree of freedom not used for encoding, Alice entangles it with the spatial modes, using polarizing beam splitters (PBS). In this way, the beam splitters (BS) reflectivities $x$, $y$, and $z$, can be effectively tuned by rotating the single-photon polarization before each PBS, using polarization controllers. We use a fast optical switch in order to select the party who performs the verification step, depending on the outcome $b$. During this operation, the photon is delayed using optical fiber spools; Alice's source together with Bob's verification setup then form a >300 m-long fibered Mach-Zehnder interferometer. In order to mitigate the resulting interference noise, we carefully insulated the spools and achieved an interference visibility of $v \gtrsim 96\%$ (see Methods for details).

Under these conditions, the thermally-induced fluctuations are slow enough such that we can easily post-select the protocol runs in which there was no phase difference between the two arms of the interferometer. This post-selection does not threaten the protocol security, as the parties could monitor the interference before performing the coin flip, and agree on starting the protocol only when the phase difference is null. Single photons are detected with threshold superconducting nanowire single-photon detectors (SNSPDs) in order

to maximize the detection efficiency. Finally, to simulate communication distance between Alice and Bob, and the corresponding losses induced by the photon storage that is necessary in this case, we use variable optical attenuators (VOAs).

Because of their central role in the analysis of the protocol, we wish to distinguish the BS reflectivities from the losses induced by the rest of the components in the setup. For that purpose we define different transmission (or heralding) efficiencies, measured when the reflectivities and the state of the switch are set to trivial values $x$, $y$, $z$, $s \in \{0, 1\}$. These values reflect the losses in every possible path in the experiment, which are induced for instance by fiber spools, VOAs, fiber coupling and mating, or detectors. We detail the notations for the efficiencies corresponding to each path and their measured values in Table 1. Each path is defined by the detector it ends in and the arm it goes through (Alice's or Bob's).

## Results with honest parties

We are first interested in the protocol when both Alice and Bob are honest. In our experiments, because of dark counts, double-pair emission, or imperfect interference visibility, Alice and Bob can still be sanctioned even though they are honest and the setup is optimized. In general, we cannot tune the reflectivities perfectly, so Alice and Bob may have slightly different winning probabilities. This means our implementation cannot satisfy perfectly the fairness and correctness conditions. Therefore, we define the fairness $\mathcal{F}$ and correctness $\mathcal{C}$ in order to quantify the closeness to these two conditions as follows:

$$\mathcal{F} = 1 - \left| \frac{\mathbb{P}_h(\text{A. wins}) - \mathbb{P}_h(\text{B. wins})}{\mathbb{P}_h(\text{A. wins}) + \mathbb{P}_h(\text{B. wins})} \right|, \tag{3}$$

$$\mathcal{C} = 1 - \frac{\mathbb{P}_h(\text{A. sanctioned}) + \mathbb{P}_h(\text{B. sanctioned})}{\mathbb{P}_h(\text{A. wins}) + \mathbb{P}_h(\text{B. wins})}. \tag{4}$$

Both quantities are equal to 1 when the corresponding conditions are perfectly fulfilled, and $\mathcal{C}, \mathcal{F} < 1$ otherwise. In our implementation, the probability of emitting a pair in a pump pulse is $p \simeq 0.015$, so double-pair emissions are highly unlikely. We condition any detection event on the detection of both a pump pulse and a heralding photon, which effectively minimizes the already low dark count rates in SNSPDs. In this way, we can omit the double-pair emissions and dark counts as a first approximation, such that only the interference visibility $v$ limits $\mathcal{C}$ and $\mathcal{F}$. Under these assumptions, we show that the

## Table 1 | List of notations and measured values for the efficiencies corresponding to the different paths involved in the experiment

| Notation | Path | $x$ | $y$ | $z$ | $s$ | Efficiency |
|---|---|---|---|---|---|---|
| $\eta_A^s$ | $x \rightarrow$ switch $\rightarrow D_A$ | 1 | | | 1 | 0.315 ± 0.008 |
| $\eta_B^y$ | $x \rightarrow y \rightarrow D_B$ | 0 | 0 | | | 0.303 ± 0.008 |
| $\eta_A^{V_1}$ | $x \rightarrow$ switch $\rightarrow z \rightarrow D_{V_1}$ | 1 | | 1 | 0 | 0.231 ± 0.008 |
| $\eta_A^{V_2}$ | $x \rightarrow$ switch $\rightarrow z \rightarrow D_{V_2}$ | 1 | | 0 | 0 | 0.219 ± 0.008 |
| $\eta_B^{V_1}$ | $x \rightarrow y \rightarrow z \rightarrow D_{V_1}$ | 0 | 1 | 0 | | 0.184 ± 0.008 |
| $\eta_B^{V_2}$ | $x \rightarrow y \rightarrow z \rightarrow D_{V_2}$ | 0 | 1 | 1 | | 0.175 ± 0.008 |

The paths are described by the PBSs (labelled by the corresponding reflectivities) and/or the switch they go through, as well as the detector at the end of the path. We also list the values of $x$, $y$, $z$, and the state of the switch $s$, required to measure these efficiencies. Values given for VOAs set at 0 dB.

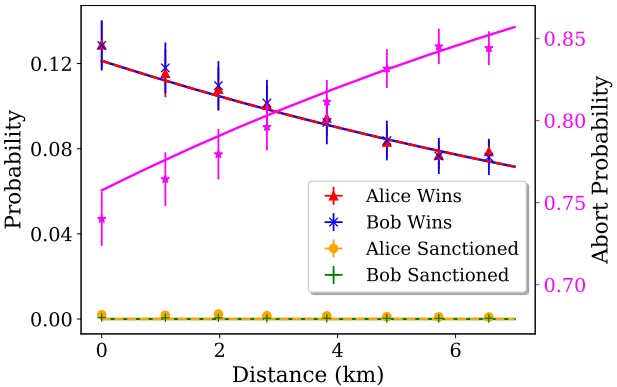

**Fig. 3 | Probability of each outcome of the protocol, measured for different communication distances between Alice and Bob.** The abort probability is shown on the right axis, in magenta. The lines represent the theoretical evolution of probabilities, calculated via Eqs. (5) to (10), with efficiencies given in Table 1. The error bars are mainly due to error propagation on these efficiencies.

correctness and fairness conditions are optimally approached by setting the following reflectivities:

$$x_h = \left[ 1 + \frac{\eta_A^{V_1}}{\eta_B^{V_1}} + \frac{\eta_A^{V_1}}{\eta_B^{y}}(1+v) \right]^{-1}, \qquad (5)$$

$$y_h = \left[ 1 + \frac{\eta_B^{V_1}}{\eta_B^{y}}(1+v) \right]^{-1}, \qquad (6)$$

$$z_h = \frac{1}{2}. \qquad (7)$$

The reader can refer to Supplementary Note 1 for the detailed proof and Supplementary Note 2 for the values used in our implementation. Then, as long as the parties are honest, we obtain the following probabilities for significant events:

$$\mathbb{P}_h(\text{A. wins}) = \mathbb{P}_h(\text{B. wins}) = x_h \eta_A^{V_1}(1+v), \qquad (8)$$

$$\mathbb{P}_h(\text{B. sanctioned}) = 0, \qquad (9)$$

$$\mathbb{P}_h(\text{A. sanctioned}) = x_h \eta_A^{V_2}(1-v). \qquad (10)$$

Note here the importance of maximizing the interference visibility $v$ so that Alice is not sanctioned while being honest. The above expressions also provide a systematic way to optimize the reflectivities for honest parties, which does not require their direct measurement (see Methods for details).

We perform the protocol for different communication distances between Alice and Bob. These are simulated by setting each of the VOAs to a transmission $\eta = e^{-0.02L}$ with $L$ the distance in kilometers, introducing additional losses to each arm of the setup. We optimize the fairness and correctness at each distance by tuning the reflectivities. We continuously run the protocol and record all detection events regardless of the phase difference between the two arms of the interferometer. Detection of both a heralding photon and a pump pulse triggers a protocol run. The average protocol rate is $\simeq 51$ kHz. As if Bob was monitoring the phase difference, we post-select the runs for which the phase spontaneously goes to zero thanks to slow temperature fluctuations, such that the rate in $D_{V_2}$ (which essentially corresponds to the probability of honest Alice being sanctioned) is

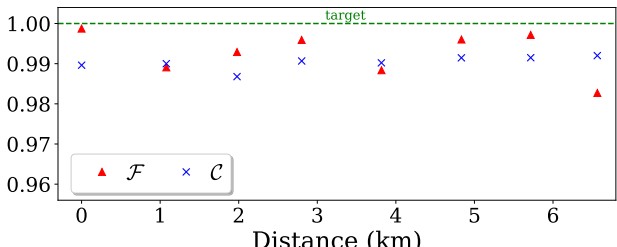

**Fig. 4 | Correctness $C$ and fairness $F$ measured in our experimental implementation of the protocol with honest parties, for different communication distances.** The dashed line gives the target value for an ideal protocol.

minimized. In this way, we measure at least $1.5 \times 10^5$ valid iterations of the protocol for a 15-min run, making the Poisson noise negligible. In Fig. 3 we give the probabilities of the different events for several distances.

We notice that the abort probability takes relatively high values, even when we trivially set the communication distance to $L = 0$ km. This has to do with important losses, particularly in mating sleeves connecting the numerous optical fiber components, the delay line, or in crystalline components such as the PBSs or the optical switch. Significant improvements could be made by fusing optical components for instance. Other critical features are the single-photon coupling and SNSPDs efficiencies. Both of these aspects are being actively studied[27–32] and could see significant improvement in the near future. We also notice that the winning probabilities of Alice and Bob are indeed very close and the probability of an honest party to be sanctioned is minimized.

To further illustrate the performance of our protocol, we show the fairness $F$ and correctness $C$ in Fig. 4. Thanks to the appropriate tuning of reflectivities $x$, $y$, and $z$, as well as low dark count rates and high visibility, we were able to keep both of these quantities very close to 1, thus approaching the ideal conditions.

### Cheat sensitivity, results with dishonest parties

The security of WCF protocols is measured by the bias that a dishonest party can induce on the flip. While we drop the balancing condition compared to the protocol in[24], our protocol retains the same security guarantees in terms of bias, i.e., the winning probability of a dishonest party is bounded away from 1 (see Supplementary Note 1). Now we highlight the cheat sensitivity of our protocol, by implementing possible attacks by dishonest parties. We consider one party to be dishonest, the other one being honest. Bob's optimal cheating strategy is quite straightforward, and consists in claiming $b = 1$ regardless of the actual measurement in detector $D_B$[24]. As Alice is honest she sets the reflectivity $x = x_h$ given in Eq. (5). When Bob claims $b = 1$ then Alice's switch directs her mode in detector $D_A$ so that she can verify whether Bob is being honest. She then detects a photon with probability:

$$\mathbb{P}(a = 1 | \text{B. cheats}) = x_h \eta_A^s, \qquad (11)$$

in which case Bob is sanctioned for cheating. Otherwise, Bob wins with probability:

$$\begin{aligned} \mathbb{P}(a = 0 | \text{B. cheats}) &= 1 - \mathbb{P}(a = 1 | \text{B. cheats}) \\ &= 1 - x_h \eta_A^s. \end{aligned} \qquad (12)$$

In this way, Alice's conditional verification, enabled in our setup by the fast optical switch, allows for a first kind of cheat sensitivity.

In order to illustrate this aspect, we implement Bob's optimal cheating strategy by systematically forcing the switch to send the photon to $D_A$. We measure the probability of sanctioning Bob for each of the communication distances simulated in the honest case. As displayed in Fig. 5, we show experimentally that the probability of

sanctioning Bob decreases as communication-induced losses increase, therefore limiting Alice's cheat sensitivity. This gives a substantial advantage to Bob when Alice's arm is particularly lossy. Note that when Bob implements that strategy, only two events are possible, namely Bob winning or Bob being sanctioned; Alice can never win except if the sanction is precisely giving Alice the win (see discussion below).

On the other hand, when Bob is honest and Alice is dishonest, her optimal cheating strategy is less straightforward. In particular, our security proof does not derive her optimal strategy but rather derives a security bound valid for all strategies (see Supplementary Note 1). Nevertheless, we can illustrate this scenario using suboptimal strategies by simply tuning the value of $x$, so that Alice sends the photon to her side with higher probability: intuitively, without taking the verification setup into account, we can naively expect Alice's winning probability to increase as she increases the reflectivity $x$. We experimentally perform the protocol for different values of $x$, all of them higher than the honest value (5). In that case, the expected event probabilities are given by the following formula (see Supplementary Note 1 for the detailed proof):

$$\mathbb{P}(\text{A. wins}) = \frac{1}{2}\left(x\eta_A^{V_1} + (1-x)y_h\eta_B^{V_1} + 2v\sqrt{x(1-x)y_h\eta_A^{V_1}\eta_B^{V_1}}\right), \quad (13)$$

$$\mathbb{P}(\text{A. sanctioned}) = \frac{1}{2}\left(x\eta_A^{V_2} + (1-x)y_h\eta_B^{V_2} - 2v\sqrt{x(1-x)y_h\eta_A^{V_2}\eta_B^{V_2}}\right), \quad (14)$$

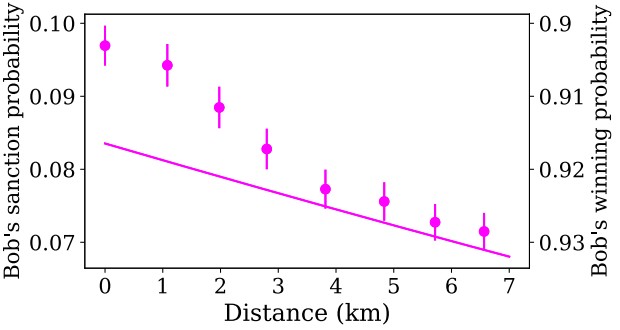

**Fig. 5 | Probabilities of Bob winning or being sanctioned, when he is performing an optimal attack, measured for different communication distances between Alice and Bob.** Only one set of points is shown for the two axes, as these two events are complementary. The line is plotted from Eqs. (11) and (12), with $\eta_A^s$ given in Table 1. The error bars are mainly due to error propagation on this efficiency. The observed deviation from the theory is linked to systematic errors when setting the reflectivities, which is discussed in Supplementary Note 2.

$$\mathbb{P}(\text{B. wins}) = (1-x)(1-y_h)\eta_B^y. \quad (15)$$

In Fig. 6(a), we show the probabilities of significant events. Contrary to our naive conjecture, we see that thanks to Bob's verification, and thus cheat sensitivity, Alice does not have a clear interest in forcing $x=1$, as her winning probability peaks around $x \simeq 0.78$.

Alice's interest in cheating actually depends on how deterrent the sanction is. We define a factor $\delta \geq 0$, which quantifies that deterrability, or alternatively how harmful the sanction is for a cheating party. From this parameter we can derive an empirical function that quantifies Alice's interest in cheating:

$$\mathcal{I}_A(\delta) = \frac{\mathbb{P}(\text{A. wins}) - \mathbb{P}(\text{B. wins}) - \delta\mathbb{P}(\text{A. sanctioned})}{\mathbb{P}(\text{A. wins}) + \mathbb{P}(\text{B. wins}) + \delta\mathbb{P}(\text{A. sanctioned})}. \quad (16)$$

This function is built such that it can be linked to the fairness (3) when taking the appropriate sanction. Indeed, if for $\delta \in [0,1]$ we sanction a cheating Alice by giving the win to Bob with probability $\delta$, then the relation $\mathcal{F} = 1 - |\mathcal{I}_A(\delta)|$ holds. In this way, $\delta = 0$ corresponds to a protocol that simply aborts without sanction when Alice is caught, and $\delta = 1$ gives a protocol that always declares Bob the winner when Alice is caught. Ultimately $\mathcal{I}_A(\delta)$ can be interpreted as a sort of expectation value of a cheating Alice, or a comparison between what she can gain by cheating and what she can lose. In Fig. 6b we plot Alice's cheating interest for different values of $\delta$ and $x$. If no sanction is taken ($\delta = 0$), we see that her interest in cheating grows with $x$. Indeed, even if her winning probability decreases for high values of $x$, Bob's then approaches zero, such that Alice wins with absolute certainty as long as the protocol does not abort. On the contrary, as the sanction is tightened and the value of $\delta$ increases, Alice has less interest in cheating for a given value of $x$. Furthermore, the value of $x$ that maximizes $\mathcal{I}_A$ also goes down, showing how strengthening the sanction actually forces Alice to adopt a strategy that leaves a chance for Bob to win. We finally discuss scenarios in which both parties cheat simultaneously in Supplementary Note 1.

## Discussion

After refining a previous theoretical proposal for a practical quantum weak coin flipping protocol[24], we were able to perform an implementation of this protocol by generating a heralded single photon, and entangling it effectively with the vacuum. Thanks to the use of low dark counts SNSPDs, tunable beam splitters and a fast optical switch, while keeping a high visibility in our fibered interferometer, we demonstrated a fair and cheat-sensitive protocol. Importantly, this last

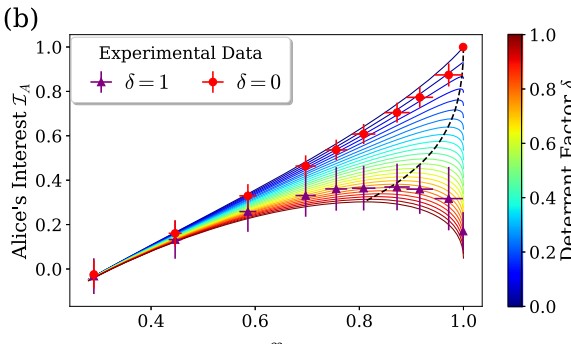

**Fig. 6 | Results for a dishonest Alice who sets different values of $x$ than the honest value.** The lines show theoretical predictions, calculated from Eqs. (13)-(16), with efficiencies given in Table 1. The error bars are mainly due to error propagation on these efficiencies. **a** Probabilities of different outcomes. **b** Alice's cheating interest for different deterrent factors $\delta$. The dashed black line indicates the points of maximum interest.

property allows to detect a cheating party with non-negligible probability.

Note that in order to sanction a dishonest party with high probability, one could systematically sanction the winning party, regardless of their honesty. Thus, in order to display genuine cheat sensitivity, we highlight the primary importance of the correctness condition, which ensures an honest party is never sanctioned for cheating. This forced us to ignore the balancing of the benefit gained by each party when adopting an optimal cheating strategy, which was previously assessed as a necessary condition for a weak coin-flipping protocol[24]. Still, we propose a way of restoring this balance, by using the deterrent factor and interest function introduced in the previous paragraphs.

The balance could indeed arise from choosing different sanctions for Alice and Bob, associated with different deterrent factors $\delta_A$ and $\delta_B$, in order to equalize the corresponding interest functions $\mathcal{I}_A(\delta_A)$ and $\mathcal{I}_B(\delta_B)$. A dishonest party who could dramatically increase their winning probability would therefore take a bigger risk of being harshly sanctioned when cheating. Interestingly enough, one could actually set arbitrarily big deterrent factors $\delta > 1$ in order to account for harsher sanctions. We leave the evaluation of these sanctions, deterrent factors and potential alternative interest functions as an interesting game theory open question.

From an experimental perspective, we remark that the robustness to losses in our implementation was illustrated by simulating communication distance with variable optical attenuators. In a practical implementation of the protocol, it would be necessary to maintain a high visibility for a longer interferometer, which could be achieved with active stabilization techniques used in twin-field quantum key distribution implementations for instance[33,34]. Furthermore, optical implementations of quantum WCF with arbitrarily small biases are yet to be discovered—such implementations would be challenging since they require a rapidly growing number of rounds of communication between the parties[35].

## Methods
### Source and detection
Our single-photon signal was heralded by its idler twin, in a pair generated via type-II SPDC in a ppKTP crystal (Raicol). We maximized the heralding efficiency $\eta_s = R_{si}/R_i$, with $R_i$ the idler photon detection rate and $R_{si}$ the pair detection rate. For that purpose, the pump focus and pair collection modes were tuned carefully when coupling to single-mode fibers, and losses on the signal-photon path were minimized. In particular, we used >85%-efficiency SNSPDs (ID281 from ID Quantique) to detect that photon. Losses on the idler photon were not limiting, so we detected it with a 25%-efficiency InGaAs APD (ID230 from ID Quantique). In this way, without adding the rest of the components, we measured a maximum heralding efficiency $\eta_s = 63\%$. All detection events were recorded by a time tagger (Time Tagger Ultra from Swabian Instruments), and dated with picosecond precision. Two detection events were considered simultaneous when measured in a 500 ps coincidence window. We also use the pump laser as a clock, in order to filter out most of the dark counts from the APD, which occur at a 1 kHz-rate. In this way, protocol runs were triggered at a rate of 51 kHz, with 40 Hz of runs mistakenly triggered by dark counts.

### Error management
Various factors can generate undesired detection events in our protocol. This is true in particular for sanction outcomes, triggered by a detection in $D_A$ or $D_{V_2}$ which should never occur when a party is honest. Most of these outcomes arise from Bob's verification procedure, which relies on a Mach–Zehnder interferometer. If this interference is of poor visibility, then $D_{V_2}$ can be triggered even if Alice is being honest, and her winning probability is also substantially lowered. Considering the length of this interferometer (>300 m), the visibility is limited by two main factors, namely the coherence length and phase fluctuations. The coherence length of photons is $\simeq 2.4$ mm, which is small enough to start losing coherence after a few hours of experiment runs. This is mostly caused by length variations in the interferometer arms due to thermal fluctuations ($\simeq 2.4$ mm/°C for a 300 m arm). We therefore regularly tune the length of one arm of the interferometer, using a free-space micrometric delay line. Phase fluctuations can be separated into two regimes. Slow phase fluctuations, of typical frequency $\lesssim 1$ Hz, are again caused by thermal variations. We can easily measure them, and then either correct them or simply post-select the desired phase differences. Fast phase fluctuations, however, are caused by noise spanning the audible spectrum from 20 Hz to 2 kHz. This noise is amplified by the 300 m fiber spools, which act as sort of microphone. These fluctuations are hard to resolve with our single-photon rate of a few 10 kHz, such that the interference pattern is averaged on that noise, and we witness an interference visibility of approximately $v \simeq 80\%$. In order to characterize that noise, we measure the interference pattern with a continuous diode laser and a fast photodiode. Without any sound insulation, the noise in the interference fluctuation spans the audible spectrum with a power spectral density of approximately $\simeq 7 \times 10^{-3} \mathrm{V}^2/\mathrm{Hz}$. In order to mitigate this noise, we wrap the fiber spools into several layers of sound-absorbing floating parquet underlay. We then achieve an interference visibility of $v \simeq 96\%$.

### Reflectivity setting
When the parties are honest, Bob first sets $z = 1/2$ by blocking Alice's signal, and equalizing the detection rates in $D_{V_1}$ and $D_{V_2}$. This later ensures an optimized interference, and therefore the correctness condition. Then he can tune $y$ such that the detection rate in $D_B$ equals twice the total rate in $D_{V_1}$ and $D_{V_2}$, which should ensure the fairness condition. Alice then tunes $x$ in order to optimize the interference visibility, which should complete the setting of reflectivities. If $v$ is significantly lower than 1, Alice and Bob might have to perform some mild adjustments on $x$ and $y$ in order to maximize the fairness and correctness. After performing a protocol with reflectivities $x, y, z$ we can evaluate them by measuring some specific probabilities (see Supplementary Note 2).

### Optical switching
During the decision step of the protocol, Bob's detection determines which party is winning, and which one is performing the verification. This decision is effectively taken into account by Alice via her optical switch (Nanospeed from Agiltron). In this way, if Bob does not claim victory, the switch is in state "0" in order to send Alice's state to Bob, who performs the verification. If Bob claims victory, the switch goes to state "1" such that Alice keeps her state and performs the verification. In practice, we send the electronic signal from Bob's detector, together with the heralding signal, to a fast programmable logic AND gate, integrated in a time controller (ID900 from ID Quantique). This AND gate filters out potential detection events outside of the protocol, which might saturate the optical switch. The gate's output signal is then sent to the optical switch, which executes the decision (see Supplementary Note 2 for more details).

## Data availability
The data that support the findings of this study are available in the Supplementary Information and from the corresponding authors upon request.

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

## Acknowledgements

We acknowledge financial support from the European Research Council project QUSCO (E.D., S.V., V.Y.), the European Commission project QUANGO (E.D.), and the PEPR integrated projects EPiQ ANR-22-PETQ-0007 (I.K., E.D.) and QCommTestbed ANR-22-PETQ-0011 (E.D.), which are part of Plan France 2030. U.C. acknowledges funding provided by the Institute for Quantum Information and Matter, an NSF Physics Frontiers Center (NSF Grant PHY-1733907). M.B. acknowledges support from the AFOSR via Q-TRUST (FA9550-21-1-0355).

## Author contributions

S.N., M.B. and E.D. designed and S.N. developed the experimental setup. S.N. and V.Y. performed the protocol implementation and processed the data. S.N., V.Y., U.C. and M.B. performed the protocol analysis. All authors discussed the analysis of the data, and contributed to writing or proofreading the manuscript. I.K. and E.D. supervised the project.

## Competing interests

The authors declare no competing interests.
