## [Peer Review File · Nature Communications]

Experimental cheat-sensitive quantum weak coin flippingREVIEWER COMMENTS

Reviewer #1 (Remarks to the Author):

In this manuscript, the authors experimentally demonstrated the quantum weak coin flipping (WCF). The protocol implemented is a refined version of their recently proposed one. The experiment shows how quantum resources can provide cheat sensitivity, which cannot be achieved by classical resources in the manner of information-theoretic security, or the so-called quantum advantage.

Overall, the results are positive as it provides a way to sense cheating which is not possible with all known classical protocols. In quantum communication, it is highly desirable to find new protocols beyond quantum key distribution for general applications. The WCF is an important cryptographic primitive. From this prospect, this work will attract general interests from both computer science and quantum information community. Because the implementation of WCF is challenging, I think this work provides the first step toward the applications of WCF. I will be very happy to recommend its publication after addressing some minor comments.

1. In page 5, the authors wrote "Both of these aspects are being actively studied and could see significant improvement in the near future." References are required before the word "and".
2. In Protocol, it seems that the outcome "0" represents the photon is not detected. I suggest the authors define it clearly in the manuscript for better understanding. Moreover, the authors mentioned detectors D_A , D_B , D_{V1} , and D_{V2} , but these detectors are not labelled in the figure inside Protocol.
3. In FIG.2, the abort probability takes relatively high values, even when the communication distance is $L = 0$ km. What is the lower bound of the abort probability when the state and measurement are perfect?
4. I'm curious in the case where both Alice and Bob are dishonest as this is the most practical assumption in WCF. But this case is not analyzed in the manuscript. Can the authors elaborate?

Reviewer #2 (Remarks to the Author):

In this paper, the authors experimentally realise, for the first time, a cheat-sensitive quantum weak coin flipping protocol, which is based on a previous proposal (by some of the authors) for an optical implementation of quantum weak coin flipping. In several aspects, the current work is an improvement compared to the original proposal, while it does leave room for further improvements.

I believe that the calculations and the results are correct, the methods used seem appropriate, and the data interpretation as well as the conclusions of the authors are well-justified and well-presented. I would like to clarify that, as my expertise is more on the theory side, if there exist issues in the implementation techniques and/or experimental details, I am not sure I would be able to spot them. The overall presentation and structure of the article are also clear.

The strong side of this work is that it is the first experimental realisation of weak coin flipping, and one of the few, so far, for quantum secure two-party computation primitives in general. That makes it an important result for this area of quantum cryptography, which is far more challenging than QKD to implement, and hopefully it will boost the interest for better implementation proposals.

The weaker point of this work is the fact that the implementation does not include Alice's optimal cheating strategy, but only suboptimal strategies. The authors roughly explain the difficulties -- both experimental and theoretical -- and they introduce a way (deterability) to somehow compensate for this. I acknowledge the difficulties as well as the effort to compensate, nevertheless this remains a relevant issue for the security, thus -- in my opinion -- rendering this work more useful for

benchmarking rather than a proposal that can stand on its own.

In any case, it would be useful to add a more detailed explanation of the experimental and theoretical challenges (instead of just referring to the previous work) and provide some perspectives for future secure implementations of coin flipping.

Overall, even though I believe this work is important as it progresses the current state of the art in the area and hopefully it will motivate more work on the field, I am not so sure about the actual extent of this progress, given the security issue from above.

Below, I suggest a few small changes that I think would improve the manuscript with respect to context.

1. It would be useful to explain the connection between cheating detection and security features; in particular, for coin flipping the security parameter is the bias of a protocol, and the security of all the previously proposed protocols is assessed by their bias (including the protocol in Ref 17). Unless their relationship is clarified, it is not easy for a reader to understand what are the security properties of the current protocol.

2. In the Introduction the authors write: "Furthermore, an explicit optical implementation of quantum WCF with arbitrarily small bias is yet to be discovered." I think, it would be fair to cite (<https://doi.org/10.1145/3357713.3384276>) and acknowledge the difficulty of such a proposal by commenting how this theoretical result affects it. Possibly, it would be better to move this part to the conclusions as future work.

3. Add a sentence with some references on experimental realisations of other primitives for secure two-party computation (e.g. <https://doi.org/10.1038/ncomms2268>, <https://doi.org/10.1038/ncomms1572>, <https://doi.org/10.1038/ncomms4717>, <https://link.aps.org/doi/10.1103/PRXQuantum.2.010335>, <https://doi.org/10.1038/ncomms4418>).

4. Cite (DOI: 10.4086/cjtcs.2016.013) together with refs 12,13.

5. Cite (arXiv:0711.4114) together with refs 10,11 and update ref 10.

Reply to the reviewers

We thank both Reviewers for their very helpful feedback, and hereby provide a reply to each of their comments/suggestions.

.....

In response to Reviewer 1:

Reviewer 1: In this manuscript, the authors experimentally demonstrated the quantum weak coin flipping (WCF). The protocol implemented is a refined version of their recently proposed one. The experiment shows how quantum resources can provide cheat sensitivity, which cannot be achieved by classical resources in the manner of information-theoretic security, or the so-called quantum advantage.

Overall, the results are positive as it provides a way to sense cheating which is not possible with all known classical protocols. In quantum communication, it is high desirable to find new protocols beyond quantum key distribution for general applications. The WCF is an important cryptographic primitive. From this prospect, this work will attract general interests from both computer science and quantum information community. Because the implementation of WCF is challenging, I think this work provides the first step toward the applications of WCF. I will be very happy to recommend its publication after addressing some minor comments.

Our reply: We thank the Reviewer for their positive assessment of our work, and for highlighting the challenging nature of our experimental demonstration of quantum weak coin flipping with quantum advantage.

Reviewer 1: 1. In page 5, the authors wrote “Both of these aspects are being actively studied and could see significant improvement in the near future.” References are required before the word “and”.

Our reply: We have added some references to recent reviews on the latest advancements of SNSPDs technology, as well as references to the latest studies demonstrating high coupling efficiencies of photons into single-mode fibers.

Reviewer 1: 2. In Protocol, it seems that the outcome “0” represents the photon is not detected. I suggest the authors define it clearly in the manuscript for better understanding. Moreover, the authors mentioned detectors D_A , D_B , D_{V_1} , and D_{V_2} , but these detectors are not labelled in the figure inside Protocol.

Our reply: We thank the Reviewer for this comment, and have changed the figure accordingly, improving the clarity: detector labels have been added, along with labels “ $b =$ ” before each path “0” and “1” after the switch. In order to avoid overloading the

figure, we have removed the outcome indices a , v_1 and v_2 , that we think are clear enough from the protocol description, and have kept the outcome b which triggers the switch. In addition, we have defined what an outcome "0" or "1" corresponds to in the Protocol box, the first time a detection occurs (at the end of step 2).

Reviewer 1: 3. In FIG.2, the abort probability takes relatively high values, even when the communication distance is $L = 0km$. What is the lower bound of the abort probability when the state and measurement are perfect?

Our reply: We thank the Reviewer for this insightful question. In the lossless scenario where all states and measurements are perfect, the abort probability is simply 0. The Reader can check this by setting all reflectivities equal to 1 and noise contributions equal to 0 in the different formulas. Intuitively, the abort outcome occurs when the photon is not detected. If the setup is lossless, then the photon has to be detected at some point, which means the protocol cannot abort. In practice, if players are honest and the interference is perfect, Bob will win if the photon is detected in D_B , or Alice will win if no photon was detected in D_B , in which case the photon will be detected in D_{V_1} .

This consideration is very relevant, as the abort probability is indeed relatively high in this first implementation, as the Reviewer noticed. However, we are confident that it could be reduced dramatically with further experimental investigations and engineering in order to mitigate the losses. As an example, we have used between 20 and 30 fiber mating-sleeves, in order to connect the different components together. Each of these mating sleeves had a transmissivity of $\approx 95\%$, therefore being responsible for a considerable portion of the losses. A fully-optimized setup for quantum network applications would surely remove such mating sleeves by fusing all components, therefore approaching the ideal case.

Reviewer 1: 4. I'm curious in the case where both Alice and Bob are dishonest as this is the most practical assumption in WCF. But this case is not analyzed in the manuscript. Can the authors elaborate?

Our reply: Most quantum two-party computation security models do not consider both parties being dishonest, as security makes sense from the perspective of an honest party willing to protect against a malicious adversary. This threat model is still very general however, as one does not make any assumption on which of the two parties is dishonest: the protocol is therefore always secure for both an honest Alice and an honest Bob.

In the case of our protocol however, understanding the double-dishonest scenario is fairly straightforward, and in fact reduces to a fully classical protocol. Since the protocol is designed in such a way that the same party (Bob) always declares the outcome of the flip first (while the verification is then performed by the losing party), Bob cannot win in any other way than declaring himself as the winner. The best that Alice can do is to then stop Bob from winning, claiming that she caught him cheating. Thus, the protocol will always abort, which is a desirable outcome in such a dishonest scenario. The case only becomes a little more complex when one considers sanctioning dishonest aborts. In

that case, Bob will always be sanctioned for cheating first, even though Alice was also dishonest.

We have added this full discussion as a Supplementary Material (A5).

In response to Reviewer 2:

Reviewer 2: In this paper, the authors experimentally realise, for the first time, a cheat-sensitive quantum weak coin flipping protocol, which is based on a previous proposal (by some of the authors) for an optical implementation of quantum weak coin flipping. In several aspects, the current work is an improvement compared to the original proposal, while it does leave room for further improvements.

I believe that the calculations and the results are correct, the methods used seem appropriate, and the data interpretation as well as the conclusions of the authors are well-justified and well-presented. I would like to clarify that, as my expertise is more on the theory side, if there exist issues in the implementation techniques and/or experimental details, I am not sure I would be able to spot them. The overall presentation and structure of the article are also clear.

The strong side of this work is that it is the first experimental realisation of weak coin flipping, and one of the few, so far, for quantum secure two-party computation primitives in general. That makes it an important result for this area of quantum cryptography, which is far more challenging than QKD to implement, and hopefully it will boost the interest for better implementation proposals.

Our reply: We thank the Reviewer for their positive assessment of our work, and for emphasizing that it is one of the few demonstrations of quantum secure two-party computations so far (generally presenting more stringent loss and noise bounds than quantum key distribution implementations).

Reviewer 2: The weaker point of this work is the fact that the implementation does not include Alice's optimal cheating strategy, but only suboptimal strategies. The authors roughly explain the difficulties – both experimental and theoretical – and they introduce a way (deterability) to somehow compensate for this. I acknowledge the difficulties as well as the effort to compensate, nevertheless this remains a relevant issue for the security, thus – in my opinion – rendering this work more useful for benchmarking rather than a proposal that can stand on its own. In any case, it would be useful to add a more detailed explanation of the experimental and theoretical challenges (instead of just referring to the previous work) and provide some perspectives for future secure implementations of coin flipping.

Overall, even though I believe this work is important as it progresses the current state of the art in the area and hopefully it will motivate more work on the field, I am not so sure about the actual extent of this progress, given the security issue from above.

Our reply: We thank the Reviewer for this important comment. While our work does

not experimentally demonstrate Alice’s optimal cheating strategy, we kindly disagree that this represents a security issue. We believe that this interpretation is due to a possibly misleading paragraph in the previous version of our manuscript.

First of all, we would like to clarify that security in our protocol is indeed measured by the bias: when we mention “*optimal*” or “*suboptimal*” strategies of a dishonest party, these are measured with respect to their cheating bias. In both cases (Alice dishonest or Bob dishonest), the security proof of the previous theoretical work from some of us (<https://doi.org/10.1103/PhysRevA.102.022414>) applies to the current protocol, with a relabelling of the experimental parameters (and dropping the balancing condition). Importantly, the security bound derived in our preceding theoretical work ensures that Alice’s cheating probability is bounded away from 1, regardless of the experimental parameters. Since this security proof is not constructive, we choose to experimentally illustrate Alice’s dishonest behavior using a suboptimal attack rather than the optimal one. However, the security guarantee of our protocol holds against any possible strategy (including infinite-dimensional ones). In particular, there is no security issue regarding the bias associated with the consideration of suboptimal strategies, since the maximal cheating bias is anyway bounded by our security bound. As such, our protocol is the first experimental implementation of secure quantum WCF and—we believe—goes beyond a simple benchmarking proposal. This constitutes **our first main contribution**.

That being said, even if the protocol is secure in terms of bias, it is only secure to some extent, i.e., a dishonest party will still have a bias advantage strictly bounded by $1/2$ compared to an honest party (note that, by one of the results which the Reviewer indicated, this will indeed be the case for all WCF protocols with a bounded number of rounds). In a practical scenario, we would thus need to incentivize an honest party to take part in such a protocol. This is precisely what we explore through cheat-sensitivity, in what constitutes **our second main contribution**. Namely, we take advantage of the natural cheat-sensitivity of our protocol and illustrate possible sanction strategies to restore balance between an honest party and a dishonest one. We do not claim to have provided an optimal solution to that problem and indeed we write: “*We leave the evaluation of these sanctions, deterrent factors and potential alternative interest functions as an interesting game theory open question*”. However, we believe that these theoretical considerations, which come on top of a rigorous security proof and are complemented by our experimental implementations of attack strategies, will motivate additional follow-up work.

In order to make the new version of our work clearer, we have added the main theoretical arguments of the security proof together with a detailed discussion in the Supplementary Information A 4, referred to in the Main Text. In particular, we added the mention that our protocol is secure regarding the bias in the introduction, and we have reformulated the following paragraph, which was potentially misleading:

“On the other hand, when Bob is honest and Alice is dishonest, her optimal cheating strategy is less straightforward, as shown in [17]. It relies on the preparation of specific states that are hard to produce with current technology. Additionally, without strong assumptions, complex methods are required in order to find such a strategy, which to this day has not been achieved yet. Still, we can perform suboptimal strategies [...]”

to “*On the other hand, when Bob is honest and Alice is dishonest, her optimal cheating strategy is less straightforward. In particular, our security proof does not derive her opti-*

mal strategy but rather derives a security bound valid for all strategies (see Supp. Mat. A). Nevertheless, we can illustrate this scenario using suboptimal strategies". We made a similar change in section A of the Supplementary Material.

Reviewer 2: Below, I suggest a few small changes that I think would improve the manuscript with respect to context.

1. It would be useful to explain the connection between cheating detection and security features; in particular, for coin flipping the security parameter is the bias of a protocol, and the security of all the previously proposed protocols is assessed by their bias (including the protocol in Ref 17). Unless their relationship is clarified, it is not easy for a reader to understand what are the security properties of the current protocol.

Our reply: We thank the Reviewer for this helpful comment. We have added the following sentence to the main text to clarify the connection between the cheating detection feature and the security properties of our protocol: *"The security of WCF protocols is measured by the bias that a dishonest party can induce on the flip. While we drop the balancing condition compared to the protocol in [17], our protocol retains the same security guarantees in terms of bias, i.e., the cheating probability of a dishonest party is bounded away from 1 (see Supp. Mat. A). Now we highlight the cheat sensitivity [...]"*

Reviewer 2: 2. In the Introduction the authors write: "Furthermore, an explicit optical implementation of quantum WCF with arbitrarily small bias is yet to be discovered." I think, it would be fair to cite (<https://doi.org/10.1145/3357713.3384276>) and acknowledge the difficulty of such a proposal by commenting how this theoretical result affects it. Possibly, it would be better to move this part to the conclusions as future work.

Our reply: We thank the Reviewer for suggesting this relevant reference, which we have included in the conclusion of the revised manuscript.

Reviewer 2: 3. Add a sentence with some references on experimental realisations of other primitives for secure two-party computation (e.g. <https://doi.org/10.1038/ncomms2268>, <https://doi.org/10.1038/ncomms1572>, <https://doi.org/10.1038/ncomms4717>, <https://link.aps.org/doi/10.1103/PRXQuantum.2.010335>, <https://doi.org/10.1038/ncomms4418>).

4. Cite (DOI: 10.4086/cjtes.2016.013) together with refs 12,13.

5. Cite (arXiv:0711.4114) together with refs 10,11 and update ref 10.

Our reply: We thank the Reviewer for these suggestions, which are now included in the revised version of our work.

.....

REVIEWERS' COMMENTS

Reviewer #1 (Remarks to the Author):

The authors have addressed my comments. I am happy to recommend its publication.

Reviewer #2 (Remarks to the Author):

The authors have addressed and clarified all the points on which I had doubts. I believe that the updated version of the manuscript can be published.